# Whole-Genome Sequencing of Linezolid-Resistant and Linezolid-Intermediate-Susceptibility *Enterococcus faecalis* Clinical Isolates in a Mexican Tertiary Care University Hospital

**DOI:** 10.3390/microorganisms13030684

**Published:** 2025-03-19

**Authors:** Pedro Martínez-Ayala, Leonardo Perales-Guerrero, Adolfo Gómez-Quiroz, Brenda Berenice Avila-Cardenas, Karen Gómez-Portilla, Edson Alberto Rea-Márquez, Violeta Cassandra Vera-Cuevas, Crisoforo Alejandro Gómez-Quiroz, Jaime Briseno-Ramírez, Judith Carolina De Arcos-Jiménez

**Affiliations:** 1HIV Unit, Hospital Civil de Guadalajara “Fray Antonio Alcalde”, Guadalajara 44280, Mexico; pedro.martinez@cucs.udg.mx; 2Health Division, Tlajomulco University Center, University of Guadalajara, Tlajomulco de Zuñiga 45641, Mexico; 3Department of Internal Medicine, Hospital Civil de Guadalajara “Fray Antonio Alcalde”, Guadalajara 44280, Mexico; leonardo.perales6284@alumnos.udg.mx (L.P.-G.); gomezportillakaren33@gmail.com (K.G.-P.); edsonreamarquez@gmail.com (E.A.R.-M.); 4Microbiology Laboratory, Hospital Civil de Guadalajara “Fray Antonio Alcalde”, Guadalajara 44280, Mexico; agomezq@hcg.gob.mx (A.G.-Q.); 2019122@mail.hcg.udg.mx (B.B.A.-C.); quigom2.0@gmail.com (C.A.G.-Q.); 5Sequencing Research and Development, Abalat, Ciudad de México 14000, Mexico; cassandra.vera@abalat.com.mx; 6Laboratory of Microbiological, Molecular and Biochemical Diagnostics (LaDiMMB), Tlajomulco University Center, University of Guadalajara, Tlajomulco de Zuñiga 45641, Mexico

**Keywords:** *Enterococcus faecalis*, linezolid resistance, *optrA*, *cfr*, multidrug-resistant bacteria, genomic surveillance

## Abstract

Linezolid-non-susceptible *Enterococcus faecalis* (LNSEf) has emerged as a critical clinical concern worldwide, yet data from Latin American settings remain scarce. This study aimed to investigate the molecular epidemiology and mechanisms underlying LNSEf in a Mexican tertiary care university hospital, focusing on clinical correlates and clonal relationships. A total of 392 non-duplicated *E. faecalis* isolates were collected over 12 months, of which 24 with minimum inhibitory concentrations ≥4 µg/mL underwent whole-genome sequencing to identify specific resistance determinants (*optrA*, *cfrA*, *23S rRNA* mutations) and to perform multilocus sequence typing (MLST) and phylogenetic analyses. Of the 392 isolates, 6.12% showed linezolid non-susceptibility, predominantly linked to plasmid- or chromosomally encoded *optrA*; only two isolates carried *cfrA*. No mutations were detected in 23S rRNA domain V or ribosomal proteins L3/L4. Clinically, LNSEf strains were associated with immunosuppression, previous surgical interventions, and prolonged hospital stays. Although most LNSEf isolates retained susceptibility to ampicillin, vancomycin, and daptomycin, they exhibited high rates of resistance to other antibiotic classes, particularly aminoglycosides and fluoroquinolones. These findings underscore the emergence of LNSEf in this region, highlighting the need for robust genomic surveillance, strict infection control, and judicious antimicrobial stewardship to curb further dissemination.

## 1. Introduction

*Enterococcus faecalis* represents a significant burden in hospital settings due to its high prevalence and associated mortality in bloodstream infections (BSIs) [1]. It is a major cause of hospital-acquired infections, particularly in patients with previous medical interventions and underlying comorbidities [2,3]. *E. faecalis* is a leading etiologic agent of BSIs and infective endocarditis, with a substantial presence in nosocomial environments [4]. Although the 30-day mortality rate for *E. faecalis* BSIs is considerable, it remains lower than that associated with *E. faecium* infections [5].

The pathogenicity of *E. faecalis* is driven by multiple virulence factors that contribute to the severity of infections in hospitalized patients [6]. Furthermore, its remarkable ability to develop antibiotic resistance poses a significant therapeutic challenge [7]. The species harbors a large set of genes that confer intrinsic resistance, enabling survival under antibiotic pressure [8]. The presence of mobile genetic elements, such as *Tn6009*, facilitates the horizontal transfer of resistance genes, promoting the spread of multidrug-resistant strains [9]. The prevalence of antibiotic-resistant *E. faecalis* varies globally but is particularly concerning in hospital settings, especially among long-term inpatients [10]

Linezolid, a key oxazolidinone antibiotic, is generally reserved as a second-line or “salvage” therapy for *Enterococcus* infections that are difficult to treat with conventional agents such as ampicillin or vancomycin [11]. Since its introduction, linezolid has been primarily used to manage infections caused by multidrug-resistant Gram-positive pathogens, including methicillin-resistant *Staphylococcus aureus* (MRSA), vancomycin-resistant enterococci (VRE), and multidrug-resistant *Mycobacterium tuberculosis* (MDR-TB) [12,13]. It is approved for the treatment of conditions such as nosocomial pneumonia, community-acquired pneumonia, complicated and uncomplicated skin and soft tissue infections, and infections caused by *Enterococcus faecium* resistant to vancomycin [12]. Additionally, its excellent oral bioavailability makes it a valuable option for step-down therapy in patients transitioning to outpatient treatment [12,14,15].

The mechanism of action of linezolid involves the inhibition of bacterial protein synthesis by binding to the 23S ribosomal RNA of the 50S subunit, thereby preventing the formation of a functional 70S initiation complex and inhibiting the translation process [16,17]. Additionally, it competes with incoming substrates at the A-site of the ribosome, located near the catalytic center, further disrupting protein synthesis [17,18]. As a result, linezolid is primarily bacteriostatic against most Gram-positive bacteria, including enterococci and staphylococci, although it can be bactericidal against certain strains of streptococci [15,19,20].

However, the emergence of linezolid-non-susceptible enterococci (LNSE) poses a significant challenge to clinical management and necessitates ongoing surveillance and a deeper understanding of resistance mechanisms [21,22,23,24]. Linezolid non-susceptibility in *Enterococcus faecalis* is relatively rare but is emerging as a concern in various regions [21,22]. For instance, a study from the United States reported a low resistance rate of 2% among *E. faecalis* isolates [21]. Similarly, a study from Iran found a linezolid non-susceptibility rate of 1.3% in *E. faecalis* isolates from urinary tract infections [22]. In contrast, a study from Korea reported a higher resistance rate of 18.8% among *E. faecalis* isolates, and in a tertiary hospital in China, the frequency of linezolid-non-susceptible *E. faecalis* was reported to be 22.61% among clinical isolates from urinary tract infections [23,24].

Epidemiological studies conducted in various regions have highlighted the sporadic nature of linezolid-resistant Enterococcus (LRE) outbreaks, suggesting that these cases often result from independent acquisition events rather than clonal dissemination [25,26]. For instance, in Chongqing, China, a study of low-level LRE strains revealed the presence of novel mutations and sequence types, indicating a diverse genetic landscape and emphasizing the importance of local surveillance [26]. Similarly, genomic epidemiology studies in China and other countries have shown the coexistence of multiple resistance mechanisms, underscoring the complexity of controlling the spread of LRE [27,28,29].

The occurrence of linezolid-non-susceptible *Enterococcus faecalis* (LNSEf) is driven by multiple mechanisms such as mutations in the 23S rRNA, alterations in ribosomal proteins L3 and L4, and the acquisition of resistance genes like *optrA*, *poxtA*, and *cfr* [21,30,31,32]. Mutations in the 23S rRNA, particularly at positions such as G2576, are well-documented mechanisms of resistance, as they directly impact the binding site of linezolid, reducing its efficacy [33,34]. Mutations in ribosomal proteins L3 and L4, although located further from the linezolid binding site, can also contribute to resistance [33,35]. These mutations may induce conformational changes in the ribosome that indirectly affect linezolid binding [35]. For instance, specific mutations in L3, such as R138V, have been associated with increased linezolid minimum inhibitory concentrations (MICs), indicating reduced susceptibility [35,36].

The *optrA* gene is a well-documented contributor to linezolid non-susceptibility, as it encodes an ATP-binding cassette (ABC) transporter that actively effluxes linezolid out of the bacterial cell, thereby reducing its intracellular concentration and effectiveness [37,38]. Studies from China have shown a high prevalence of *optrA* in various settings, including hospital sewage and community environments, often associated with other antimicrobial resistance genes (*ARGs*), such as *fexA* and *erm(A)*, which can co-transfer with *optrA* via mobile genetic elements like plasmids and transposons [27,38,39].

The *poxtA* gene, although less frequently discussed, also contributes to linezolid non-susceptibility by encoding a protein that protects the ribosome from the action of oxazolidinones [40]. It has been identified in various *Enterococcus* species, including *E. faecalis*, and is often found on plasmids, facilitating its horizontal transfer [40,41]. The presence of *poxtA*, along with *optrA*, can lead to higher levels of resistance due to their combined effects on linezolid susceptibility [40,41,42].

The *cfr* gene affects linezolid susceptibility by encoding a methyltransferase that modifies the 23S rRNA at position A2503 [43]. This methylation alters the binding site of linezolid on the bacterial ribosome, thereby reducing the drug’s ability to inhibit protein synthesis [44]. The *cfr* gene confers a multidrug-resistant phenotype, not only to oxazolidinones like linezolid but also to phenicols, lincosamides, pleuromutilins, and streptogramin A compounds [43]. The *cfr* gene is often located on mobile genetic elements such as plasmids and transposons, facilitating its horizontal transfer across different bacterial species and contributing to the dissemination of resistance [45].

The regional variability in the prevalence of linezolid non-susceptibility and associated resistance mechanisms highlights the need for robust local and regional surveillance efforts. Although *E. faecalis* with linezolid non-susceptibility remains relatively uncommon in many areas, its emergence as a clinical concern necessitates continuous monitoring. Characterizing the genetic basis of resistance and elucidating the clonal relationships among isolates can provide valuable insights into the local epidemiology of linezolid-non-susceptible enterococci while contributing to the global understanding of this evolving threat.

In this context, the present study aimed to investigate the molecular epidemiology and resistance mechanisms of *E. faecalis* strains isolated from a Mexican tertiary care university hospital. By mapping the distribution of resistance determinants and analyzing clonal lineages, we sought to generate actionable insights to inform infection control strategies and optimize therapeutic approaches for this emerging pathogen.

## 2. Materials and Methods

### 2.1. Bacterial Strain Collection and Antimicrobial Susceptibility Testing

A total of 392 non-duplicated *Enterococcus faecalis* strains were obtained in the microbiology laboratory of our institution from June 2023 to June 2024. The strains were collected from various sample types, including tissue, urine, blood culture, surgical wound secretion, abscess, peritoneal fluid, bone, and cerebrospinal fluid (CSF). Upon receipt, specimens were cultured on blood agar (BD Diagnostics, Bergen, NJ, USA) and incubated at 35–37 °C for 24–48 h under aerobic conditions. Presumptive colonies with morphological and hemolytic patterns consistent with Enterococcus were subjected to initial phenotypic tests. These included Gram staining (to confirm Gram-positive cocci), catalase testing (to differentiate from staphylococci), and the bile esculin test.

All presumptive *Enterococcus* isolates were definitively identified using the VITEK^®^ 2 system (BioMérieux, Lyon, France). Colonies from overnight culture plates were suspended in 0.45% saline to the turbidity range recommended by the manufacturer, and the suspensions were loaded into the VITEK^®^ 2 instrument for processing. Antimicrobial susceptibility testing (AST) was performed using the AST-GP75 and P663 cards (BioMérieux) according to the manufacturer’s instructions and using strict quality control procedures. The antibiotics tested included ampicillin (AMP), ciprofloxacin (CIP), daptomycin (DAP), nitrofurantoin (NIT), gentamicin (GEN), streptomycin (STR), levofloxacin (LEV), linezolid (LNZ), benzylpenicillin (PEN), tetracycline (TET), and vancomycin (VA). Results were interpreted following the Clinical and Laboratory Standards Institute (CLSI) guidelines [46], with specific emphasis on linezolid susceptibility in *Enterococcus faecalis* (LSEf) at ≤2 µg/mL, intermediate resistance at 4 µg/mL, and resistance at ≥8 µg/mL. For the purposes of this study, isolates displaying intermediate or resistant profiles (≥4 µg/mL) were collectively classified as linezolid-non-susceptible *E. faecalis* (LNSEf) [46].

### 2.2. Clinical Data Collection and Case–Control Selection

Clinical records of patients with linezolid-non-susceptible *E. faecalis* isolates were thoroughly reviewed. Sociodemographic, microbiological, and clinical data were systematically collected, including the following: age, gender, admission diagnosis, occupation, prior antibiotic use within the preceding 90 days and 14 days (along with specific antibiotic types), previous hospitalizations, comorbidities, the Charlson comorbidity index, surgical history within the past three months (including the type of surgery, if applicable), the use of vasopressor agents, Intensive Care Unit (ICU) admission, initial empirical treatment, the length of the hospital stay, and in-hospital mortality. To identify factors associated with LNSEf, each patient with an LNSEf isolate was matched by age, gender, and clinical ward to two patients with linezolid-susceptible *E. faecalis* isolates. This case–control approach enabled a focused comparison of risk factors and outcomes between linezolid-non-susceptible *Enterococcus faecalis* cases and their matched controls.

### 2.3. DNA Extraction, Library Construction, and Whole-Genome Sequencing

Genomic DNA was extracted from pure cultures of *Enterococcus faecalis* using the ZymoBIOMICS DNA Miniprep Kit (Zymo Research, Irvine, CA, USA), following the manufacturer’s instructions. The purity of the DNA was assessed by spectrophotometry (Nanodrop One C, Thermo Fisher Scientific, Waltham, MA, USA), selecting samples with an A260/280 ratio between 1.8 and 2.0. The concentration of DNA was quantified using fluorometry (Qubit 4.0, Invitrogen, Thermo Fisher Scientific, Waltham, MA, USA) with the dsDNA High-Sensitivity (HS) Assay. Extracted DNA was subsequently stored at −70 °C.

For whole-genome library preparation, the Illumina DNA Prep protocol (Illumina, San Diego, CA, USA) was used, which applied a bead-based transposome complex to fragment genomic DNA. The process was automated using the Biomek NGeniuS Next Generation Library Prep System (Beckman Coulter, Brea, CA, USA), with a total DNA input of 20 ng and programmed using 8 cycles to amplify tagmented DNA using IDT indexes set A. Library quality was assessed using the Standard (S2) Cartridge Kit with a Qsep1 instrument (BioOptic Inc., New Taipei City, Taiwan), which resulted in fragment sizes of approximately 600–700 bp. The concentration and dilution of DNA were further verified using the dsDNA HS Assay fluorometry (Qubit 4.0, Invitrogen, Thermo Fisher Scientific, Waltham, MA, USA). The prepared libraries were sequenced on an Illumina NextSeq 2K platform (Illumina, San Diego, CA, USA) in paired-end mode with 150 bp read lengths. Flow cell loading and sequencing were performed following Illumina protocols to ensure data quality and integrity.

### 2.4. Bioinformatic Analyses

The bioinformatic workflow was directed toward the genomic characterization of *Enterococcus faecalis*, emphasizing molecular typing, the detection of antimicrobial resistance (AMR) genes—particularly for linezolid non-susceptibility—and phylogenetic analysis.

Initial quality control (QC) of the sequencing data in FASTQ format, representing clinical isolates of *E. faecalis*, was performed using FastQC (v0.11.9 Babraham Bioinformatics, Cambridge, UK) and MultiQC (v1.14 Seqera Labs, Barcelona, Spain) [47,48]. These tools evaluated read quality metrics such as per-base sequence quality, GC content, and adapter contamination, ensuring that the data met quality standards for downstream analyses (Appendix A). Taxonomic classification was conducted using Kraken2 (v2.1.2 Johns Hopkins University, Baltimore, MD, USA), with a reference database enabling the isolation of *E. faecalis*-specific reads [49]. Non-target sequences were discarded, ensuring that the subsequent steps focused exclusively on the target organism. To enhance the quality of the data, trimmed reads were generated using fastp (v0.23.2 Shenzhen, Guangdong, China) [50]. This process removed low-quality bases and adapter sequences, resulting in high-quality datasets optimized for genome assembly.

Genome assembly was performed with SPAdes (v4.1.0 San Petersburgo, Russia) using the “careful” mode to reduce assembly errors [51]. The resulting assemblies were evaluated using QUAST (v5.3 San Petersburgo, Russia), with metrics including the number of contigs, total genome length (~2.8–3 Mbp for *E. faecalis*), N50, and GC content (~37%) (Appendix A) [52]. These metrics provided an assessment of the assembly completeness and contiguity.

The raw sequences were made publicly available via the Zenodo platform at https://doi.org/10.5281/zenodo.14873829. The genome assemblies of the 24 LNSEf generated in this study were deposited in GenBank under BioProject ID PRJNA1217060, with individual genome accessions JBLVLD000000000–JBLVKX000000000 (Appendix A).

Antimicrobial resistance gene detection efforts centered on identifying resistance determinants, including genes associated with linezolid non-susceptibility (*optrA*, *cfr*, *poxtA*) and other antimicrobial classes. To accomplish this, we employed (ARIBA v2.14.6 Cambridgeshire, UK) with the ResFinder (v4.1 Kongens Lyngby, Denmark) and CARD (version 4.0.0 databases Hamilton, ON, Canada), enabling the direct detection of AMR genes from both raw reads and assembled genomes [53]. Additionally, LRE-Finder (version 1.0.0 Copenhague, Dinamarca) was employed to identify *optrA*, *cfr*, *cfrB*, and *poxtA* genes, as well as common mutations in the V domain of the 23S rRNA (G2576T and G2505A) in LNSEf isolates [54].

All detected *optrA* sequences were blasted against the complete *optrA* gene sequence from plasmid pE349 (GenBank Accession No. NG_048023.1). Specific mutations associated with linezolid non-susceptibility were further analyzed using BWA-MEM (Cambridge, MA, USA) for read mapping and bcftools for variant calling, ensuring the precise detection of resistance-related genetic variations [55,56]. To predict additional linezolid resistance genes such as *cfrD*, the Resistance Gene Identifier (RGI) version 6.0.3 (Hamilton, Ontario, Canada) was used, utilizing reference data from CARD (version 4.0.0) [57].

To validate the identity of the detected resistance genes, all sequences corresponding to *optrA*, *cfr*, and related genes were translated into amino acid sequences using Prodigal (version 2.6.3). These protein sequences were subsequently subjected to BLASTP DIAMOND, (version 2.1.10 Baden-Württemberg, Germany) analysis against the non-redundant protein (NR) database of the NCBI, with stringent screening parameters of ≥99% sequence identity and an E-value threshold of <1 × 10^−10^ [58]. This approach ensured high-confidence matches and confirmed the presence of functional resistance determinants.

The localization of *optrA* genes was determined by performing BLASTN searches against the NCBI nt database using contigs containing *optrA* sequences. Plasmid-specific replicons were detected using PlasmidFinder (version 2.1, database 4 December 2023 Kongens Lyngby, Dinamarca), and alignment results were analyzed to compare identity and coverage with known plasmid and chromosomal sequences [59].

Assembled contigs were analyzed utilizing the ABRicate tool (version 1.0.1; University of Melbourne, Melbourne, Australia) to identify virulence-associated genes. The virulence factor database (VFDB; Beijing, China) was employed with a minimum identity threshold of 80% and a minimum gene coverage of 80%. Virulence genes were identified based on their sequence similarity to previously reported entries in the VFDB [60].

Molecular typing was performed using multilocus sequence typing (MLST) via the PubMLST database (https://pubmlst.org/organisms/enterococcus-faecalis, accessed on 22 January 2025), classifying isolates into sequence types (STs) based on the allelic profiles of seven housekeeping genes (*gdh*, *gyd*, *pstS*, *gki*, *aroE*, *xpt*, and *yqiL*).

Phylogenetic analysis for each *Enterococcus faecalis* isolates involved extracting core-genome single-nucleotide polymorphisms (SNPs) using Parsnp (v2.1.2; Johns Hopkins University, Baltimore, MD, USA), followed by SNP alignments generated with vcf2phylip (v2.0; Munich, Germany), and the construction of maximum-likelihood (ML) phylogenetic trees with IQ-TREE (v2.4.0; Amsterdam, Netherlands) using the GTR+G model and 1000 ultrafast bootstrap replicates [61,62,63]. For *optrA* phylogenetic analysis, sequencing reads were aligned to the *optrA* reference sequence (NG_048023.1) using Bowtie2 (v2.5.1; Johns Hopkins University, Baltimore, MD, USA), and variants were identified with Snippy (v4.6.0; University of Melbourne, Melbourne, Australia), retaining only high-confidence SNPs (QUAL > 100, DP ≥ 10). Consensus sequences from each isolate were aligned with MAFFT (v7.526; Osaka University, Osaka, Japan), and an ML phylogenetic tree was reconstructed using IQ-TREE (v2.4.0, GTR + G model, 1000 bootstrap replicates). Trees were visualized in iTOL (v6.8; Heidelberg, Germany), elucidating evolutionary relationships and strain clustering [63,64,65,66,67].

### 2.5. Statistical Analysis

Demographic data are reported as simple relative frequencies. The normality of the data distribution was assessed via the Shapiro–Wilk test. Pearson’s chi-square test and Fisher’s exact test were used to compare proportions, as appropriate. For comparisons of quantitative variables, Student’s t tests and Wilcoxon–Mann–Whitney tests were used for normally and nonnormally distributed data, respectively.

Statistical analyses were conducted using Python (v3.9; Wilmington, DE, USA) for data processing and visualization, leveraging libraries such as pandas (v1.3.5), NumPy (v1.21.4), and Matplotlib (v3.5.0). Additionally, R software (v4.2.2; Vienna, Austria) was utilized for statistical analyses, validation, and graphical representation.

## 3. Results

### 3.1. Study Population and Sociodemographic and Clinical Characteristics

Of the 392 non-duplicated *Enterococcus faecalis* strains, we identified 26 isolates with linezolid MICs ≥ 4 µg/mL through automated phenotypic methods, which were subsequently subjected to whole-genome sequencing (WGS). After excluding isolates with inadequate readings or insufficient quality for downstream analysis, 24 isolates were selected for molecular typing, the characterization of linezolid-non-susceptibility mechanisms, and phylogenetic tree construction. Additionally, 12 linezolid-susceptible isolates underwent WGS to assess genetic diversity, of which 3 were excluded due to insufficient quality for further analysis.

Finally, 48 linezolid-susceptible isolates were documented from clinical records (including the 12 previously described isolates that underwent WGS) for sociodemographic and clinical comparisons. The study workflow, which encompassed whole-genome sequencing, antimicrobial resistance characterization, and the sociodemographic and clinical profiling of patients with linezolid-susceptible and -non-susceptible *Enterococcus faecalis* isolates, is depicted in Figure 1.

Among the 24 patients (7 women, 17 men) with linezolid-non-susceptible *E. faecalis* isolates, the median age was 48 years (interquartile range [IQR]: 31–57). Comorbidities were identified in 66.7% of patients (n = 16), with a median Charlson comorbidity index of 1 (IQR: 0–3). The most common comorbidities included diabetes mellitus (25.0%), hypertension (20.83%), immunosuppression (12.50%), and chronic kidney disease (8.33%). Overall, 66.7% of patients had undergone surgical intervention, while 12.5% required admission to the ICU. The median hospital stay was 14 days (IQR: 11.25–20.25), and the in-hospital mortality was 16.7% (n = 4).

In terms of occupation, unemployed individuals represented the most common category, accounting for seven cases (29.17%). Housewives comprised five cases (20.83%), followed by students with four cases (16.67%). Construction workers were affected in two cases (8.33%), while delivery workers, commerce workers, engineers, and couriers each accounted for one case (4.17%).

Sex-based comparisons revealed that women had a higher median Charlson comorbidity index (3 vs. 0; *p* = 0.059) and a significantly higher in-hospital mortality rate (75% vs. 25%; *p* = 0.040) than men. Conversely, men were significantly more likely to have undergone recent surgical intervention (87.5% vs. 12.5%; *p* = 0.045). These findings highlight notable sex-related differences in comorbidity burden and clinical outcomes among patients with linezolid-non-susceptible *E. faecalis*.

With respect to prior antibiotic exposure, 45.83% (n = 11) of patients had received antibiotic treatment within the past 90 days, while 37.5% (n = 9) had recent antibiotic exposure within the two weeks preceding sample collection. Regarding the specific antibiotics administered before resistance was detected, cefazolin plus gentamicin (CZ + GEN) and ciprofloxacin (CIP) were the most frequently used, each in 8.33% (n = 2) of cases. Other antibiotics were used in single cases, each accounting for 4.17% of cases, including ceftriaxone plus clindamycin plus ciprofloxacin (CRO + CLI + CIP), ceftriaxone (CRO), meropenem plus ceftriaxone (MEM + CRO), ceftriaxone plus clindamycin (CRO + CLI), ceftriaxone plus metronidazole plus amikacin (CRO + MTZ + AN), cefotaxime (CTX), and ceftazidime plus vancomycin (CAZ + VA). The remaining clinical and sociodemographic characteristics of linezolid-non-susceptible *E. faecalis* isolates are detailed in Table 1.

In the matched comparison of 24 patients with linezolid-non-susceptible *E. faecalis* and 48 patients with linezolid-susceptible *E. faecalis*—matched by age, sex, and hospital department—the LNSEf group exhibited significantly higher rates of immunosuppression (12.5% vs. 0%; *p* = 0.0339) and prior surgical interventions (66.7% vs. 22.9%; *p* = 0.0008), along with a significantly longer median hospital stay (14 days [IQR: 11.25–20.25] vs. 9.5 days [IQR: 3.75–15]; *p* = 0.02).

Although prior hospitalization within the preceding 90 days (54.2% vs. 31.2%; *p* = 0.104) and the presence of any comorbidity (70.8% vs. 47.9%; *p* = 0.111) were more frequent in the LNSEf group, these differences did not reach statistical significance. Notably, in-hospital mortality rates were identical in both groups (16.7%; *p* = 1).

Overall, these findings suggest that LNSEf infection was associated with a greater burden of immunosuppression, a higher frequency of surgical interventions, and prolonged hospitalization, while other sociodemographic and clinical factors remain largely comparable. Further details are presented in Table 2.

The documented cases of LNSEf were distributed across various hospital departments. Traumatology, General Surgery, and Thorax and Cardiovascular Surgery each accounted for 12.5% (n = 3). The Cardiology and Adult Infectious Diseases departments represented 8.3% (n = 2), with most cases concentrated in surgical units.

Regarding sample sources, urine was the most common site of isolation (29.2%; n = 7). Surgical wound exudates, foot tissue samples, and blood cultures from central lines each accounted for 8.3% (n = 2). The sociodemographic and clinical characteristics of patients with LNSEf are detailed in Table 3, while an extended complementary version is available in Appendix A.

### 3.2. Antimicrobial Susceptibility Profiles

With respect to the antimicrobial susceptibility profile, ampicillin, penicillin, nitrofurantoin, and vancomycin exhibited complete (100%) susceptibility. Similarly, daptomycin demonstrated 100% susceptibility according to Clinical and Laboratory Standards Institute (CLSI) criteria (MIC ≤ 4 µg/mL). In contrast, tetracycline resistance was notably high (91.7%), and significant rates of high-level aminoglycoside resistance were observed for gentamicin (66.7%) and streptomycin (50%). Levofloxacin resistance affected 62.5% of isolates, with an additional 4.2% displaying intermediate MIC values, while ciprofloxacin resistance reached 66.7%. The phenotypic and genotypic characteristics, along with hospital distribution, are presented in Table 3, and the overall resistance profile is illustrated in Appendix A.

### 3.3. Molecular Typing and Genotypic Analysis of Linezolid Resistance Mechanisms

Genotypic analysis identified 15 distinct sequence types among the 24 LNSEf isolates. The most frequent was ST585 (16.7%; n = 4), followed by ST32 (12.5%; n = 3). Two isolates (8.3%) were assigned to each of ST506 and ST476. The remaining STs—ST101, ST283, ST287, ST202, ST480, ST376, ST179, ST16, ST69, ST415, and ST40—each accounted for 4.2% (n = 1). Additionally, two isolates (8.3%) were classified as ‘Not Determined’ (ND).

The primary resistance mechanism was the *optrA* gene, present in all isolates. The *cfrA* gene, co-occurring with *optrA*, was detected in two isolates (8.3%). The most frequent *optrA* variants were G1879A, C1933T (45.8%) and T411G, T626G, G866A (29.2%). A smaller subset (8.3%) carried the T10G, T35A, C54T, A91G, A107G, T626G, C949T, A1966G profile. Two isolates exhibited an even more complex mutation set (including A134T, G1278A, A1331G, A1541C, C1933T) together with *cfrA*. No mutations were identified in 23S rRNA (G2505A, G2576U) or ribosomal proteins *rplC* (L3) and *rplD* (L4).

The analysis using BLASTN identified *optrA* in both chromosomal and plasmidic locations, with 11 isolates (45.8%) in each. In two isolates (8.3%), *optrA* was chromosomal while *cfrA* was plasmidic. The presence of *optrA* on plasmids suggests a high potential for horizontal transfer, whereas chromosomal integration may indicate stable inheritance within specific *E. faecalis* genetic backgrounds. To validate resistance determinants, *optrA* and *cfrA* protein sequences underwent BLASTP (DIAMOND) analysis against the NCBI non-redundant protein database. High sequence identity (≥99%) and significant E-values (<1 × 10^−10^) confirmed their classification as functional ABC-F type ribosomal protection proteins. The identification of antibiotic resistance proteins using BLASTP analysis in clinical isolates is presented in Appendix A.

In addition to *optrA* and *cfrA*, a variety of antimicrobial resistance genes were identified. The most prevalent were *lsa(A)* and *fexA* (100%), followed by *erm(B)* (95.8%) and *erm(A)* (87.5%), both associated with macrolide resistance. The *dfrG* gene (87.5%) conferred resistance to trimethoprim. The aminoglycoside-modifying enzymes *ant(6)-Ia* and *aph(3′)-III* were detected in 83.3% of isolates, while *tet(L)* (75.0%) and *tet(M)* (8.3%) contributed to tetracycline resistance. Other notable resistance determinants included *aac(6′)-aph(2″)* (58.3%), *cat* (45.8%), *lnu(B)* (37.5%), and fluoroquinolone resistance-associated mutations in *parC* (S80I) and *gyrA* (S83I). These findings underscore a multidrug-resistant profile that may significantly limit treatment options.

Analysis using PlasmidFinder revealed a diverse set of plasmid replicons among the 24 LNSEF isolates. The most frequently detected replicon was *repUS43*, identified in 16 isolates. Other commonly found plasmid types included *rep9b* (five isolates), *rep7a* (three isolates), and *rep1* and *rep9c*, each present in two isolates. Less frequently detected replicons included *rep2*, *rep6*, *rep22*, and *rep18b*, each observed in a single isolate. Notably, four isolates did not present any detectable plasmid replicons. The presence of multiple plasmid replicons suggests a potential role in the dissemination of antimicrobial resistance genes within *Enterococcus faecalis* populations.

Further analysis revealed that certain STs exhibited consistent *optrA* variants with uniform genomic localization in the isolates examined. For instance, both ST101 and ST202 carried the T10G, T35A, C54T, A91G, A107G, T626G, C949T, A1966G variant exclusively on plasmids, while ST16 and ST283 displayed the T626G, A1541C variant, also on plasmids. In contrast, isolates classified as ST179 consistently harbored the G1879A, C1933T variant integrated into the chromosome. Notably, however, G1879A, C1933T appeared in multiple ST backgrounds and could be chromosomal or plasmid-borne in different isolates, highlighting the overall plasticity of *optrA*. These findings suggest that certain STs appear predisposed to harbor specific *optrA* variants in particular genomic contexts, which may influence the stability and transmissibility of linezolid resistance within *Enterococcus faecalis* populations.

Most *E. faecalis* isolates shared a core set of virulence factors involved in capsule formation (*cps* genes), biofilm production (pilus-associated ebp genes and *srtC*), and proteolytic activity (*fsrA/B/C*, gelE, sprE). In certain sequence types (e.g., ST585, ST179), the presence of cytolysin genes (*cylA/B/M/S/I/R*) further enhanced their virulence potential. Additionally, elements such as *asa1* and *prgB/asc10* suggested an increased capacity for aggregation and plasmid transfer. Collectively, these findings underscore the multifaceted pathogenicity of *E. faecalis*, where adhesion, biofilm formation, exoenzyme production, and, in specific STs, cytolysin-mediated toxicity play key roles in its virulence. The complete set of virulence factors is presented in Appendix A.

### 3.4. Phylogenetic Analysis of Enterococcus Clinical Isolates

A phylogenetic tree was constructed using whole-genome sequences to explore the evolutionary relationships among the 24 LNSEf strains and 9 LSEf strains. The analysis revealed a diverse phylogenetic structure, with LNSEF isolates distributed across multiple sequence types and hospital departments, suggesting widespread dissemination. The sequence types ST32, ST585, and ST476 were the most prominent, whereas ST101 was identified in only one isolate, confirming that it was not a major lineage among the LNSEf isolates. Linezolid-non-susceptible isolates were scattered across different clades, rather than clustering into a single lineage, suggesting the independent emergence of resistance across diverse genetic backgrounds. The presence of the *optrA* gene was consistently associated with linezolid non-susceptibility, with two isolates also harboring *cfrA*, suggesting potential horizontal gene transfer. Interestingly, ST32 exhibited two distinct *optrA* variants: one plasmid-borne (T411G, T626G, G866A), and another chromosomally integrated (T10G, T35A, C54T, A91G, A107G, A134T, T626G, C949T, G1278A, A1331G, A1541C, C1933T), the latter accompanied by *cfrA* in a plasmid. Additionally, *optrA* remained chromosomal in all ST585 isolates, whereas ST506 and ST476 consistently carried *optrA* in the chromosome as well. These findings suggest that specific *optrA* variants may be preferentially associated with either chromosomal or plasmidic integration, depending on the genetic background. The phylogenetic structure highlights a concerning trend of increasing non-susceptible isolates from late 2023 to mid-2024, likely driven by selective pressure. Additionally, specific *optrA* mutations (G1879A, C1933T, A1441C, T411G) were detected in distinct STs, supporting the hypothesis of the parallel evolution of resistance mechanisms. The association between phylogenetic distribution, resistance profiles, and hospital departments where isolates were identified is presented in Figure 2.

Finally, a phylogenetic tree was constructed using *optrA* sequences to investigate the evolutionary relationships among LNSEF isolates (Figure 3). The analysis revealed a structured yet diverse distribution of *optrA* sequences, with distinct clusters corresponding to specific STs and genomic localizations. Different *optrA* variants evolved independently across multiple genetic backgrounds rather than from a single ancestral strain. Notably, chromosomal *optrA* variants formed distinct clades separate from plasmid-borne ones, reinforcing the role of genomic context in shaping linezolid non-susceptibility. Isolates belonging to ST585 and ST506 consistently harbored chromosomal *optrA*, clustering together despite originating from different hospital units. In contrast, those classified as ST32 and ST476 exhibited a higher frequency of plasmid-associated *optrA*, suggesting an increased potential for horizontal gene transfer. Some isolates exhibited both chromosomal and plasmidic *optrA* variants, particularly within ST32, indicating recombination events or parallel resistance acquisition. This aligns with *cfrA* co-occurrence in *optrA*-harboring plasmids, supporting mobile genetic element-mediated dissemination. The diversity of *optrA* sequences also correlated with functional polymorphisms, with mutations (G1879A, C1933T, A1441C, T411G, T626G) clustering within specific STs. These mutations may modulate linezolid resistance levels, contributing to different phenotypic expressions of non-susceptibility. Overall, the phylogenetic structure highlights the complexity of *optrA* evolution in *Enterococcus faecalis*, with multiple independent acquisition events shaping resistance in the hospital setting. The relationship between *optrA* sequence diversity, genetic background, and resistance mechanisms is detailed in Figure 3.

## 4. Discussion

This study provides critical insights into the prevalence and mechanisms of linezolid non-susceptibility among *Enterococcus faecalis* isolates in a Mexican tertiary care university hospital. Among the 392 isolates analyzed, 6.12% (n = 24) exhibited linezolid non-susceptibility, including 2.8% (n = 11) with intermediate susceptibility and 3.32% (n = 13) classified as resistant, primarily mediated by the *optrA* gene.

According to the medical literature, linezolid resistance in *Enterococcus faecalis* is generally low but not negligible, with geographically variable prevalence [21,22,23,24,37,38,68,69,70]. Although our prevalence was higher than that reported in some developed countries, it has not yet reached the levels observed in certain regions of South Korea or China [23,24].

Historically, the primary mechanism of linezolid resistance across multiple species (e.g., *Staphylococcus aureus*, *Enterococcus faecium*) has been mutations in the domain V region of the 23S rRNA, most commonly G2576T in staphylococci [71,72,73,74]. However, in *E. faecalis*, the plasmid-borne *optrA* gene has emerged in recent years as the predominant driver of linezolid nonsusceptibility in many geographic regions [22,23,24,25,26].

Compared to other linezolid resistance mechanisms, such as point mutations in the 23S rRNA or the presence of *cfr*, the *optrA* gene provides a significant advantage due to its frequent localization on mobile genetic elements (plasmids or transposons), facilitating horizontal transfer between strains and even across different *Enterococcus* species [37]. This mobility allows *optrA* to disseminate rapidly in both clinical and community settings, often co-occurring with additional resistance genes (e.g., *fexA*) on the same plasmid, thereby expanding the multidrug resistance profile [22,23,24,25,26]. While ribosomal mutations remain chromosomal and non-transferable, and *cfr*, despite its potential for plasmid localization, appears less prevalent than *optrA* in *E. faecalis* isolates worldwide, *optrA* has emerged as the dominant reservoir for linezolid resistance [22,23,24,25,26].

Consistent with this trend, our data revealed a lower prevalence of *cfrA*-mediated resistance, with only two isolates carrying this gene, in contrast to the higher rates reported in studies from China and South Korea [23,24]. Additionally, the absence of 23S rRNA mutations differed from findings in European cohorts, suggesting regional variability in resistance mechanisms [75]. No mutations were identified in ribosomal proteins L3 and L4 or in the *poxtA* gene among our isolates. These findings underscore the predominance of alternative resistance mechanisms and emphasize the need for ongoing surveillance.

Clinical analysis revealed that LNSEF isolates were associated with higher rates of immunosuppression, prior surgical history, and prolonged hospital stays. This aligns with the medical literature, which identifies risk factors for linezolid-resistant enterococcal infections, including prior exposure to linezolid and other antibiotics, prolonged hospitalization, and underlying conditions such as gastrointestinal surgery, urogenital disorders, malignancies, diabetes, and polymicrobial infections [25,76]. These factors create selective pressure, facilitating the emergence and persistence of resistant strains [25,76].

However, none of the patients in our cohort had documented prior linezolid use, suggesting that resistance may spread through mechanisms beyond direct selective pressure from this agent. This finding is consistent with reports of the horizontal gene transfer of linezolid resistance determinants (e.g., *optrA*) among enterococci, as well as potential cross-resistance arising from prior exposure to other antibiotics [76]. The presence of resistant strains in patients without prior linezolid exposure underscores the need for stringent infection control measures and highlights the potential for resistant clones to circulate within healthcare environments, driven by multiple overlapping risk factors.

The phylogenetic analysis confirmed significant genetic diversity among LNSEF isolates, with 19 distinct STs identified. Notably, certain STs exhibited consistent *optrA* variants and genomic localization. ST101 and ST202 harbored *optrA* variants exclusively on plasmids, while ST16 and ST283 carried *optrA* on plasmids with identical mutations (*T626G*, *A1541C*). In contrast, ST179 consistently exhibited the *G1879A*, *C1933T optrA* variant in the chromosome. These findings suggest that specific *optrA* variants may be preferentially associated with either chromosomal or plasmidic integration, potentially influencing the stability and dissemination of linezolid resistance.

Additionally, whole-genome sequencing revealed that LNSEF isolates were widely distributed across hospital departments, supporting both clonal expansion and independent resistance acquisition. The clustering of ST32, ST585, ST101, and ST16 in multiple departments suggests potential nosocomial transmission and highlights the need for targeted infection control strategies.

The identification of diverse sequence types, including ST585 and ST32, supports the polyclonal dissemination of LNSEF within our institution, consistent with reports of *Enterococcus faecalis* resistant to linezolid-harboring *optrA* and *cfr* genes in hospital settings [77]. This widespread polyclonal distribution highlights the complexity of managing antibiotic resistance, as it facilitates the persistence and transmission of resistant strains across diverse bacterial populations and hospital environments [78]. The ability of *E. faecalis* to disseminate resistance through multiple genetic backgrounds poses a significant challenge for infection control, emphasizing the urgent need for enhanced surveillance and containment strategies.

The interplay between *optrA* polymorphisms, their genetic context (plasmid vs. chromosome), and the co-occurrence of additional resistance determinants, such as *cfrA*, collectively modulates the degree of linezolid resistance [79,80]. Our findings indicate that the presence of *optrA* alone does not uniformly confer high-level resistance; instead, resistance levels may be influenced by specific mutations within *optrA,* its genomic location, and the strain’s background. Isolates harboring *optrA* exhibit MIC values ranging from 4 µg/mL (intermediate) to ≥8 µg/mL (resistant), consistent with reports that plasmid-borne *optrA* generally correlates with higher MICs [81]. However, emerging evidence suggests that certain chromosomal variants can also confer resistance ≥8 µg/mL, likely through additional genetic factors or specific sequence types that enhance gene expression [68].

Beyond reduced susceptibility to linezolid, our *E. faecalis* isolates exhibited extensive resistance to multiple antibiotic classes. High-level aminoglycoside resistance (HLAR), characterized by impaired synergy with gentamicin or streptomycin, correlated with the presence of *aac(6′)-aph(2″)*, *ant(6)-Ia*, and *aph(3′)-III* genes. Fluoroquinolone resistance was associated with *gyrA* (S83I) and/or *parC* (S80I) mutations, while the near-universal carriage of *tet(L)* and/or *tet(M)* explained the widespread resistance to tetracyclines. Additional resistance determinants, including *erm(B)*, *cat*, *fexA*, and *fosB*, were identified as conferring resistance to macrolides, phenicols, and fosfomycin. Despite this extensive multidrug resistance profile, most isolates remained susceptible to ampicillin, penicillin, vancomycin, and daptomycin, reinforcing their continued role as viable therapeutic options. This observation aligns with prior reports showing that *E. faecalis* isolates carrying linezolid resistance determinants—such as *optrA* or *cfr*—frequently retain susceptibility to β-lactams, glycopeptides, and daptomycin [37,39,80]. The mechanisms underlying linezolid resistance (often ribosomal target modifications or efflux) do not typically confer cross-resistance to these alternative antibiotic classes, underscoring the importance of comprehensive susceptibility testing to guide effective treatment choices.

Several limitations should be acknowledged. First, during data collection, some variables were missing and had to be excluded from the analyses. While strict inclusion criteria were applied to ensure data quality, the retrospective nature of clinical and sociodemographic data collection inherently limits the ability to control for all potential confounders. Prospective studies are needed to further evaluate the robustness of factors associated with LNSEf acquisition.

Second, confirmatory methods such as broth microdilution or E-tests were not incorporated, limiting the ability to assess resistance levels beyond MICs > 8 µg/mL. However, all isolates with linezolid MICs ≥ 4 µg/mL harbored genetic determinants associated with resistance. Additionally, it should be noted that current EUCAST clinical breakpoints define linezolid resistance as MIC ≥ 4 µg/mL [82]. Therefore, based on these criteria, all isolates included in this study could be considered linezolid-resistant.

Finally, the single-center design of this study, along with the relatively small number of sequenced isolates (n = 24), may limit the generalizability of our findings, particularly in settings with different patient demographics, healthcare infrastructures, or antimicrobial resistance patterns.

Our findings emphasize the critical role of genomic surveillance and antimicrobial stewardship programs, particularly in high-risk hospital settings. The identification of novel *optrA* mutations offers opportunities for further research into their functional implications and potential as therapeutic targets. Future multicenter studies are needed to validate these observations and capture broader epidemiological trends.

Integrating genomic surveillance into routine clinical practice is essential to monitoring and controlling the spread of LNSEf. Strengthening surveillance efforts, implementing robust infection control measures, and optimizing antimicrobial stewardship strategies will be key to limiting the dissemination of linezolid-non-susceptible *E. faecalis* within healthcare settings.

## 5. Conclusions

In conclusion, our findings highlight the emerging clinical significance of linezolid-non-susceptible *Enterococcus faecalis* (LNSEf) within a Mexican tertiary care hospital, where 6.12% of *E. faecalis* isolates exhibited intermediate or full resistance to linezolid, primarily driven by *optrA*. The notable sequence type (ST) diversity, along with the presence of both plasmid- and chromosomally encoded *optrA* variants—occasionally co-occurring with *cfr*—reflects the complex evolutionary trajectories of these resistant strains. The absence of prior linezolid exposure in most patients underscores the potential for horizontal gene transfer and highlights the need for enhanced infection control measures. Clinically, LNSEf isolates were more frequently identified in patients with immunosuppression, recent surgical interventions, or prolonged hospital stays, suggesting a multifactorial risk profile beyond direct linezolid usage. Although most LNSEf isolates remained susceptible to ampicillin, vancomycin, and daptomycin, the co-occurrence of additional resistance determinants—such as *aac(6′)-aph(2″)*, *erm(B)*, and *tet(L)*—illustrates a broader multidrug-resistant phenotype. These findings underscore the importance of ongoing genomic surveillance, antimicrobial stewardship, and targeted infection control strategies to curb LNSEf dissemination and preserve the efficacy of critical antimicrobials. Reinforcing hand hygiene and contact precautions, promoting judicious antibiotic use, and integrating routine whole-genome sequencing into epidemiological surveillance will be key steps in mitigating LNSEf transmission and protecting vulnerable patient populations.

## Figures and Tables

**Figure 1 microorganisms-13-00684-f001:**
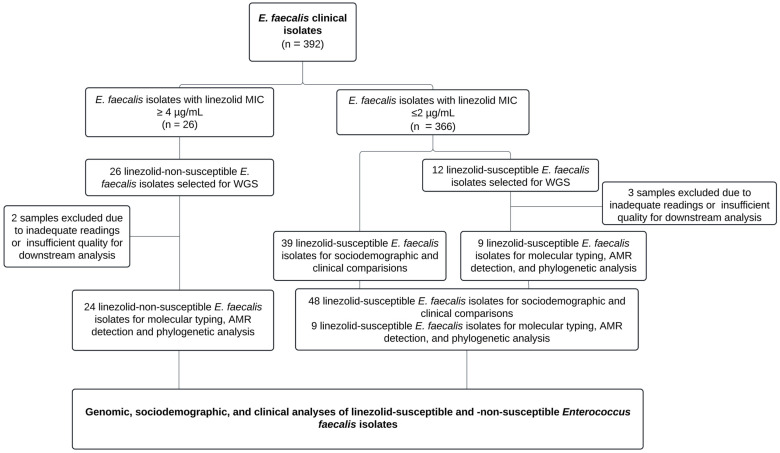
Study workflow for whole-genome sequencing and characterization of *Enterococcus faecalis* isolates.

**Figure 2 microorganisms-13-00684-f002:**
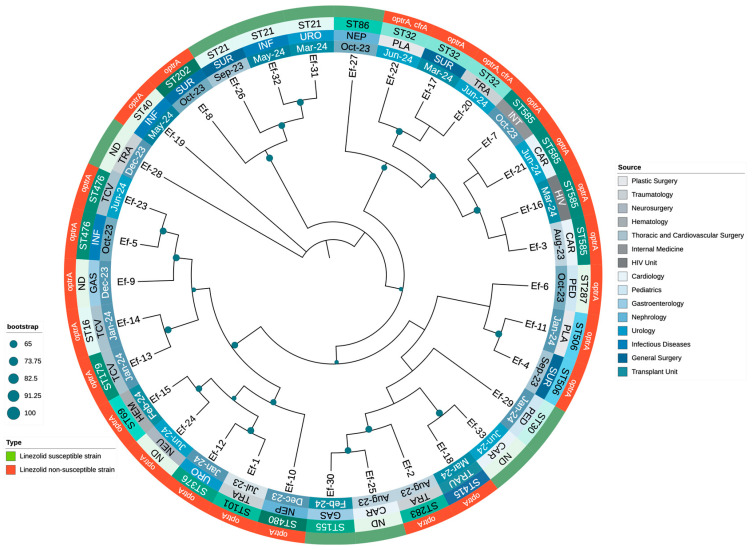
Phylogenetic distribution and clinical sources of LNSEf and LSEf isolates throughout the study period.

**Figure 3 microorganisms-13-00684-f003:**
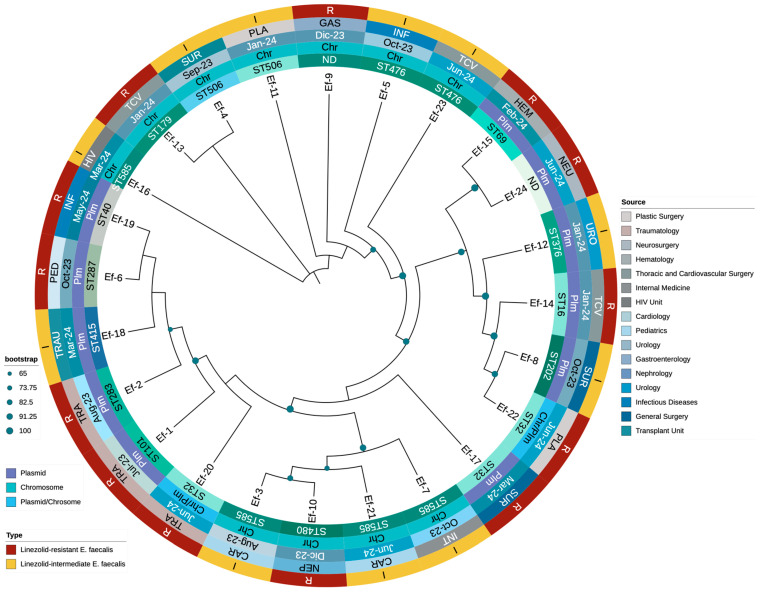
Phylogenetic distribution of LNSEf isolates based on *optrA* sequences, illustrating their genomic location (plasmidic or chromosomal), source of isolation, and corresponding linezolid susceptibility.

**Table 1 microorganisms-13-00684-t001:** Sociodemographic and clinical characteristics of patients with linezolid-non-susceptible *Enterococcus faecalis* isolates distributed by sex.

Variable	Total (n = 24)	Women (%) (n = 7)	Men (%) (n = 17)	*p* Value
Age—median (IQR)	48 (31.0–57.0)	56 (53.5–62.25)	43 (27.0–55.0)	0.092
Previous antibiotic use in 90 days—n (%)	11 (45.8)	3 (27.3)	8 (72.7)	1
Previous antibiotic in 14 days—n (%)	9 (37.5)	3 (33.3)	6 (66.7)	0.643
Previous hospitalization in 90 days—n (%)	13 (54.2)	4 (30.8)	9 (69.2)	0.66
Comorbidities—n (%)	16 (66.7)	6 (37.5)	10 (62.5)	0.1243
Charlson comorbidity index—median (IQR)	1 (0–3)	3 (1.5–3.75)	0 (0–2)	0.059
Diabetes mellitus—n (%)	6 (25.0)	3 (50)	3 (50)	0.2786
Hypertension—n (%)	5 (20.8)	3 (60)	2 (40)	0.0886
Cardiovascular disease—n (%)	0 (0)	0 (0)	0 (0)	1
Obesity—n (%)	1 (4.2)	1 (100)	0 (0)	0.260
Immunosuppression—n (%)	3 (12.5)	1 (33.3)	2 (66.7)	1
Chronic kidney disease—n (%)	2 (8.3)	0 (0)	2 (100)	1
Surgical Intervention—n (%)	16 (66.7)	2 (12.5)	14 (87.5)	0.045
Central line insertion—n (%)	9 (37.5)	3 (33.3)	6 (66.7)	0.643
Urinary catheter—n (%)	16 (66.7)	5 (31.2)	11 (68.8)	0.621
Need for vasopressors—n (%)	7 (29.2)	3 (42.9)	4 (57.1)	0.318
ICU admission—n (%)	3 (12.5)	2 (66.7)	1 (33.3)	0.155
Hospital stay—median (IQR)	14 (11.3–20.3)	14 (10.3–14.8)	15 (12.0–23.0)	0.343
In-hospital mortality—n (%)	4 (16.7)	3 (75.0)	1 (25.0)	0.040

**Table 2 microorganisms-13-00684-t002:** Sociodemographic and clinical characteristics of patients with linezolid-susceptible and linezolid-non-susceptible *Enterococcus faecalis* isolates.

Variable	Total (n = 72)	LSEf (n = 48)	LNSEf (n = 24)	*p* Value
Age—median (IQR)	48 (31.25–57)	48 (31.25–57.25)	48 (31–57)	0.9096
Male sex—n (%)	51 (70.8)	34 (66.7)	17 (70.8)	0.928
Previous antibiotic use in 90 days—n (%)	34 (47.2)	23 (47.9)	11 (45.8)	1
Previous Antibiotic in 14 days—n (%)	29 (40.3)	20 (41.7)	9 (37.5)	0.932
Previous hospitalization in 90 days—n (%)	28 (38.9)	15 (31.2)	13 (54.2)	0.104
Comorbidities—n (%)	40 (55.6)	23 (47.9)	17 (70.8)	0.111
Charlson comorbidity index—median (IQR)	1 (0–3)	1.5 (0–3)	1 (0–3)	0.905
Diabetes mellitus—n (%)	15 (20.8)	9 (19.6)	6 (25.0)	0.826
Hypertension—n (%)	15 (20.8)	10 (20.8)	5 (20.8)	1
Cardiovascular disease—n (%)	0 (0)	0 (0)	0 (0)	1
Obesity—n (%)	2 (2.8)	1 (2.1)	1 (4.2)	1
Immunosuppression—n (%)	3 (4.2)	0 (0)	3 (12.5)	0.033
Chronic kidney disease—n (%)	10 (13.9)	8 (16.7)	2 (8.3)	0.478
Surgical intervention—n (%)	27 (37.5)	11 (22.9)	16 (66.7)	<0.001
Central line insertion—n (%)	25 (34.8)	15 (31.2)	10 (41.7)	0.540
Urinary catheter—n (%)	40 (55.6)	23 (47.9)	17 (70.8)	0.111
Need for vasopressors—n (%)	14 (19.4)	7 (14.6)	7 (29.2)	0.246
ICU admission—n (%)	11 (15.3)	8 (16.7)	3 (13.0)	1
Hospital stay—median (IQR)	12 (4–17.25)	9.5 (3.75–15.0)	14 (11.25–20.25)	0.02
In-hospital mortality—n (%)	12 (16.7)	8 (16.7)	4 (16.7)	1

**Table 3 microorganisms-13-00684-t003:** Phenotypic and genotypic profiles of 24 linezolid-non-susceptible *Enterococcus faecalis* isolates.

Label	Date	Source	AMP	CIP	DAP	NIT	HLG	HLS	LEV	LNZ	PEN	TET	VA	MLST	Genes Associated with Linezolid Resistance	Locations of Linezolid Resistance Genes	Other AMR Genes	Other Identified Plasmids
Ef-1	23 July	Traumatology	≤2	1	4	≤16	SYN-S	SYN-S	1	≥8	2	≥16	1	ST101	*optrA* (T10G, T35A, C54T, A91G, A107G, T626G, C949T, A1966G)	Plasmid	*erm(B)*, *fexA*, *lsa(A)*, *NarA*, *NarB*, *tet(L)*	*repUS43*
Ef-2	23 August	Traumatology	≤2	1	1	≤16	SYN-S	SYN-R	1	≥8	2	≥16	≤0.5	ST283	*optrA* (T626G, A1541C)	Plasmid	*ant(6)-Ia*, *aph(2″)-Ic*, *aph(3′)-III*, *dfrG*, *erm(A)*, *erm(B)*, *fexA*, *lsa(A)*, *tet(L)*, *tet(M)*	*rep2*, *rep6*, *rep9b*
Ef-3	23 August	Cardiology	≤2	≥8	2	≤16	SYN-R	SYN-R	≥8	4	2	≥16	1	ST585	*optrA* (G1879A, C1933T)	Chromosome	*aac(6′)-aph(2″)*, *ant(6)-Ia*, *ant(6)-Ia*, *ant(9)-Ia*, *aph(3′)-III*, *dfrG*, *erm(A)*, *erm(B)*, *fexA*, *lnu(B)*, *lsa(A)*, *lsa(E)*,, *parC (S80I)*, *str*, *tet(L)*	*rep7a*, *repUS43*
Ef-4	23 September	General Surgery	≤2	1	4	≤16	SYN-S	SYN-S	1	4	1	≥16	1	ST506	*optrA* (G1879A, C1933T)	Chromosome	*ant(9)-Ia*, *aph(3′)-III*, *dfrG*, *erm(A)*, *erm(B)*, *fexA*, *lsa(A)*, *tet(L)*	-
Ef-5	23 October	Infectious Diseases	≤2	≥8	2	≤16	SYN-R	SYN-S	≥8	4	2	≤1	1	ST476	*optrA* (G1879A, C1933T)	Chromosome	*aac(6′)-aph(2″)*, *ant(6)-Ia*, *ant(9)-Ia*, *aph(3′)-III*, *dfrG*, *erm(A)*, *erm(B)*, *fexA*, *gyrA (S83I)*, *lsa(A)*, *parC (S80I*	-
Ef-6	23 October	Pediatrics	≤2	4	2	≤16	SYN-R	SYN-R	4	≥8	2	≥16	1	ST287	*optrA* (G1879A, C1933T)	Plasmid	*aac(6′)-aph(2″)*, *ant(6)-Ia*, *aph(3′)-III*, *dfrG*, *erm(A)*, *erm(B)*, *fexA*, *lsa(A)*, *NarA*, *NarB*, *tet(L)*	*rep1*, *rep9b*
Ef-7	23 October	Internal Medicine	≤2	≥8	1	≤16	SYN-R	SYN-S	≥8	4	2	≥16	1	ST585	*optrA* (G1879A, C1933T)	Chromosome	*aac(6′)-aph(2″)*, *ant(6)-Ia*, *ant(9)-Ia*, *aph(3′)-III*, *cat*, *dfrG*, *erm(A)*, *erm(B)*, *fexA*, *lsa(A)*, *parC (S80I)*, *tet(L)*	*repUS43*
Ef-8	23 October	General Surgery	≤2	4	2	≤16	SYN-S	SYN-S	4	4	2	≥16	1	ST202	*optrA* (T10G, T35A, C54T, A91G, A107G, T626G, C949T, A1966G)	Plasmid	*ant(9)-Ia*, *dfrG*, *erm(B)*, *erm(B)*, *fexA*, *fosB*, *lsa(A)*, *NarA*, *NarB*, *tet(L)*	-
Ef-9	23 December	Gastroenterology	≤2	≥8	2	≤16	SYN-R	SYN-S	≥8	≥8	1	≥16	1	ND	*optrA* (G1879A, C1933T)	Chromosome	*aac(6′)-aph(2″)*, *ant(6)-Ia*, *ant(9)-Ia*, *aph(3′)-III*, *cat*, *dfrG*, *erm(A)*, *erm(B)*, *fexA*, *fosB*, *gyrA (S83I)*, *lsa(A)*, *NarA*, *NarB*, *parC (S80I)*, *tet(L)*	-
Ef-10	23 December	Nephrology	≤2	≥8	2	≤16	SYN-S	SYN-S	≥8	≥8	2	≥16	1	ST480	*optrA (*(G1879A, C1933T)	Chromosome	*ant(9)-Ia*, *aph(3′)-III*, *dfrG*, *erm(A)*, *erm(B)*, *fexA*, *lsa(A)*, *NarA*, *NarB*, *tet(L)*	*repUS43*
Ef-11	24 January	Plastic Surgery	≤2	1	2	≤16	SYN-R	SYN-R	1	4	1	≥16	≤0.5	ST506	*optrA* (G1879A, C1933T)	Chromosome	*aac(6′)-aph(2″)*, *ant(6)-Ia*, *ant(6)-Ia*, *ant(9)-Ia*, *aph(3′)-III*, *cat*, *dfrG*, *erm(A)*, *erm(B)*, *fexA*, *lnu(B)*, *lsa(A)*, *lsa(E)*, *tet(L)*	*repUS43*
Ef-12	24 January	Urology	≤2	4	2	≤16	SYN-R	SYN-R	2	4	8	≥16	1	ST376	*optrA* (T411G, T626G, G866A)	Plasmid	*aac(6′)-aph(2″)*, *ant(6)-Ia*, *ant(6)-Ia*, *aph(3′)-III*, *cat*, *dfrG*, *erm(A)*, *fexA*, *lnu(B)*, *lsa(A)*, *lsa(E)*, *tet(L)*	*repUS43*
Ef-13	24 January	Thoracic and Cardiovascular Surgery	≤2	≤0.5	4	≤16	SYN-R	SYN-R	1	≥8	2	≥16	1	ST179	*optrA* (G1879A, C1933T)	Chromosome	*aac(6′)-aph(2″)*, *ant(6)-Ia*, *ant(9)-Ia*, *aph(3′)-III*, *cat*, *erm(A)*, *erm(B)*, *fexA*, *lnu(B)*, *lsa(A)*, *lsa(E)*, *tet(M)*	*repUS43*
Ef-14	24 January	Thoracic and Cardiovascular Surgery	≤2	≤0.5	2	≤16	SYN-R	SYN-R	1	≥8	2	≥16	1	ST16	*optrA* (T626G, A1541C)	Plasmid	*aac(6′)-aph(2″)*, *aph(3′)-III*, *dfrG*, *erm(A)*, *erm(B)*, *fexA*, *lnu(B)*, *lsa(A)*, *lsa(E)*, *tet(M)c*	*rep9b*, *repUS43*
Ef-15	24 February	Hematology	≤2	≥8	2	≤16	SYN-S	SYN-S	≥8	≥8	2	≥16	≤0.5	ST69	*optrA* (T411G, T626G, G866A)	Plasmid	*cat*, *dfrG*, *erm(B)*, *fexA*, *lsa(A)*, *parC*	*rep22*, *repUS43*
Ef-16	24 March	HIV Unit	≤2	≥8	1	≤16	SYN-R	SYN-R	≥8	4	2	≥16	≤0.5	ST585	*optrA* (G1879A, C1933T)	Chromosome	*aac(6′)-aph(2″)*, *ant(6)-Ia*, *ant(6)-Ia*, *ant(9)-Ia*, *aph(3′)-III*, *cat*, *dfrG*, *erm(A)*, *erm(B)*, *fexA*, *lnu(B)*, *lsa(A)*, *lsa(E)*, *parC (S80I)*, *str*, *tet(L)*	*rep7a*, *repUS43*
Ef-17	24 March	General Surgery	≤2	≥8	4	≤16	SYN-S	SYN-S	≥8	≥8	2	≥16	≤0.5	ST32	*optrA* (T411G, T626G, G866A)	Plasmid	*ant(6)-Ia*, *aph(3′)-III*, *erm(A)*, *erm(B)*, *fexA*, *gyrA (S83I)*, *lsa(A)*, *parC (S80I)*, *tet(L)*	*repUS43*
Ef-18	24 March	Transplant Unit	≤2	4	2	≤16	SYN-S	SYN-S	4	4	2	≥16	2	ST415	*optrA* (T411G, T626G, G866A)	Plasmid	*cat*, *dfrG*, *erm(A)*, *erm(B)*, *fexA*, *fosB3*, *lsa(A)*, *NarA*, *NarB*, *tet(L)*	*rep1*, *rep9c*
Ef-19	24 May	Infectious Diseases	≤2	≤0.5	2	≤16	SYN-R	SYN-R	0.5	≥8	2	≥16	1	ST40	*optrA* (T411G, T626G, G866A)	Plasmid	*aac(6′)-aph(2″)*, *ant(6)-Ia*, *ant(6)-Ia*, *aph(3′)-III*, *dfrG*, *erm(A)*, *erm(B)*, *fexA*, *lnu(B)*, *lsa(A)*, *lsa(E)*, *tet(L)*	*rep18b*, *rep9b*, *repUS43*
Ef-20	24 June	Traumatology	≤2	4	2	≤16	SYN-R	SYN-R	≥8	≥8	2	≥16	1	ST32	*optrA* (T10G, T35A, C54T, A91G, A107G, A134T, T626G, C949T, G1278A, A1331G, A1541C, C1933T), *cfrA*	*optrA*, chromosome; *cfrA*, plasmid	*aadD*, *ant(9)-Ia*, *aph(2″)-Ic*, *aph(3′)-III*, *bleO*, *cat*, *dfrG*, *erm(A)*, *erm(B)*, *fexA*, *fosB3*, *lnu(A)*, *lsa(A)*, *tet(L)*	*repUS43*
Ef-21	24 June	Cardiology	≤2	≥8	1	≤16	SYN-R	SYN-S	≥8	4	2	≥16	1	ST585	*optrA* (G1879A, C1933T)	Chromosome	*aac(6′)-aph(2″)*, *ant(6)-Ia*, *ant(9)-Ia*, *aph(3′)-III*, *cat*, *dfrG*, *erm(A)*, *fexA*, *lnu(B)*, *lsa(A)*, *lsa(E)*, *parC (S80I)*, *str*, *tet(L)*	*rep7a*, *repUS43*
Ef-22	24 June	Plastic Surgery	≤2	4	2	≤16	SYN-R	SYN-R	≥8	≥8	2	≥16	1	ST32	*optrA* (T10G, T35A, C54T, A91G, A107G, A134T, T626G, C949T, G1278A, A1331G, A1541C, C1933T)*, cfrA*	*optrA*, chromosome; *cfrA*, plasmid	*aadD*, *ant(9)-Ia*, *aph(2″)-Ic*, *aph(3′)-III*, *bleO*, *cat*, *dfrG*, *erm(A)*, *erm(B)*, *fexA*, *fosB3*, *lnu(A)*, *lsa(A)*, *tet(L)*	*repUS43*
Ef-23	24 June	Thoracic and Cardiovascular Surgery	≤2	≥8	2	≤16	SYN-R	SYN-S	≥8	4	2	≤1	1	ST476	*optrA* (T411G, T626G, G866A)	Chromosome	*aac(6′)-aph(2″)*, *ant(6)-Ia*, *ant(9)-Ia*, *aph(3′)-III*, *dfrG*, *erm(A)*, *erm(B)*, *fexA*, *gyrA (S83I)*, *lnu(B)*, *lsa(A)*, *lsa(E)*, *parC (S80I)*	*rep9b*, *rep9c*
Ef-24	24 June	Neurosurgery	≤2	≤0.5	2	≤16	SYN-R	SYN-R	0.5	≥8	2	≥16	1	ND	*optrA* (T411G, T626G, G866A)	Plasmid	*aac(6′)-aph(2″)*, *ant(6)-Ia*, *ant(9)-Ia*, *aph(3′)-III*, *dfrG*, *erm(A)*, *erm(B)*, *fexA*, *lsa(A)*, *parC (S80I)*, *tet(L)*	*repUS43*

## Data Availability

The original data presented in this study are openly available through the Zenodo platform at https://doi.org/10.5281/zenodo.14873829. The genome assemblies of the 24 LNSEf strains generated in this study were deposited in GenBank under BioProject ID PRJNA1217060, with individual genome accession numbers ranging from JBLVLD000000000 to JBLVKX000000000. Additionally, further documentation can be found in the Appendix A.

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
