# Peer review of "Whole-Genome Sequencing of Linezolid-Resistant and Linezolid-Intermediate-Susceptibility Enterococcus faecalis Clinical Isolates in a Mexican Tertiary Care University Hospital"

_microorganisms, 2025, doi:10.3390/microorganisms13030684_

Round 1
Reviewer 1 Report
Comments and Suggestions for Authors
The authors investigated the susceptibility of 392 strains of E. faecalis collected over 12 months in a Mexican hospital to detect the linezolid resistance mechanisms of E. faecalis. From all samples, the authors retained 6.12% of strains, for which the linezolid minimum inhibitory concentrations were ≥4 µg/mL. These were subjected to whole-genome sequencing. The results obtained reveal that, of the 392 isolates, 6.12% demonstrated linezolid non-susceptibility, predominantly associated with plasmid- or chromosomally encoded. The strains analysed were susceptible to ampicillin, vancomycin, and daptomycin. However, most linezolid-non-susceptible Enterococcus faecalis isolates (LNSEf) possessed resistance to other classes of antibiotics, particularly aminoglycosides and fluoroquinolones.
The studies performed concluded that LNSEf poses a growing threat in the studied region. This fact imposes the need for genomic surveillance, antimicrobial stewardship, and targeted infection control strategies to curb the dissemination of linezolid-non-susceptible enterococci and preserve the efficacy of critical antimicrobials, thereby limiting the spread of these strains.
In my opinion, this is the novelty of this study, and for this reason, I recommend its publication.
The manuscript is well written, with results presented concisely and logically, and the conclusions are supported by the results. The subject treated in this manuscript fits with the journal's aims.
However, before publishing, minor revisions are still needed, as follows:
1)Before the ''Materials and Methods'' section, at the end of the ''Introduction'' section, the authors must clearly state the aim of their study.
2) A sentence cannot begin with an abbreviation. The authors must carefully read their manuscript and revise these sentences (see the sentences beginning at the following rows: R151; R162; R192).
3) Table 1 is not mentioned in the manuscript text.
4) At row 438, instead of "Figure 2", the authors must write "Figure 3".
5) For the figure presented between rows 238-439, the authors must specify what it represents (e.g., Figure 3...).
Author Response
We sincerely thank the reviewers for their time and effort in evaluating our manuscript. Their suggestions and recommendations have significantly improved the quality of this work. As part of this round of revisions, we have incorporated the requested corrections.
Attached to this platform, we provide a detailed description of the changes made, which have been highlighted in the revised manuscript for easy identification.

Reviewer 2 Report
Comments and Suggestions for Authors
- In Table S1, add n, the absolute number of samples.
- It is also worth adding to the discussion about EUCAST, which has a slightly different interpretation than CLSI. S ≤ 4, R >4.
The disadvantage of the manuscript is the relatively small number of sequenced samples (24). The great advantage of the manuscript is the comparison of sequencing results with clinical data and data on susceptibility to other antibiotics, as well as the analysis of not only chromosomal DNA, but also plasmid DNA. The methods are appropriate and relevant.
Author Response

(The authors gave the same response as above.)

Reviewer 3 Report
Comments and Suggestions for Authors
The study explored the whole-genome sequencing of linezolid-resistant and linezolid-intermediate susceptibility enterococcus faecalis clinical isolates in a Mexican tertiary care university hospital. When reviewing the paper, consider the following points:
- In the abstract section, please clearly state the purpose of the study.
- In the introduction section, it is necessary to clarify the main problem in the study, which is the Enterococcus faecalismicrobe, what it is, what are its problems and dangers, and how widely linezolid is used to treat it?
- In the Materials and Methods section, the collection and identification of bacterial strains, collection of clinical data, and antibiotic susceptibility testing are combined into one paragraph, while each would be better described in separate paragraphs with more details.
- I wonder why informed consent was not obtained from the patients whose clinical records were relied upon?
- The entire Materials and Methods section lacks references and citations! Are these the authors' methods!?
- Line 414, Figure 2 and line 438 also Figure 2! This figure is untitled.
- Adding subheadings in the results section makes it easier for the reader to follow along.
- Several published research have focused on the identification and genetic characterisation of linezolid-resistant Enterococcus faecalisin clinical isolates from various regions throughout the world. Citing these research in the discussion section can help to enrich it.
- line 568, “Clinically, LNSEf isolates were more frequently identified in patients with immunosuppression, recent surgical interventions, or prolonged hospital stays, suggesting a multifactorial risk profile beyond direct linezolid usage”, in light of these results, are there any specific recommendations for the study to reduce the risks to public health?

Author Response

(The authors gave the same response as above.)

Round 2
Reviewer 3 Report
Comments and Suggestions for Authors
The manuscript has been reviewed, taking into account the proposed amendments and necessary corrections.